# Discourses on Positive Animal Welfare by Sheep Farmers and Industry Actors: Implications for Science and Communication

**DOI:** 10.3390/vetsci11100452

**Published:** 2024-09-24

**Authors:** Mukhtar Muhammad, Jessica Elizabeth Stokes, Louise Manning, Iona Yuelu Huang

**Affiliations:** 1Department of Agriculture, Science and Practice, Royal Agricultural University, Cirencester GL7 6JS, UK; jessica.stokes@rau.ac.uk; 2Lincoln Institute for Agri-Food Technology, University of Lincoln, Riseholme Park, Lincoln LN2 2LG, UK; lmanning@lincoln.ac.uk; 3Harper Adams Business School, Harper Adams University, Newport TF10 8NB, UK; ihuang@harper-adams.ac.uk

**Keywords:** positive animal welfare, awareness, interpretation, social identity, knowledge dissemination, anthropomorphism, self-identity, language, qualitative research

## Abstract

**Simple Summary:**

This research looked at how sheep farmers and industry actors in the UK understand, think about, and define “positive animal welfare”. It, therefore, involved interviewing farmers and industry experts, including veterinarians and advisors. The study found that how the farmers (n = 25) and industry actors (n = 11) perceived and viewed what positive animal welfare is differed from those perceptions held by scientists. Some of the study participants linked positive welfare to “positive stockmanship” and “good animal welfare” frames, while others proposed broader ideas and meanings, linked to existing scientific positive welfare definitions, including considering “high welfare” states and “happy, healthy” dimensions. Overall, the findings suggested that scientists should work closely with farmers and industry actors to bridge the gap between academic discourse and perceived meanings held by the farmers and by the industry, to develop practical and effective methods for understand how stakeholders define this concept, and the adoption of positive animal welfare approaches and practices.

**Abstract:**

This research examines how sheep farmers and industry actors in the United Kingdom (UK) understand and conceptualize what animal welfare scientists term ‘positive animal welfare’. It explores their awareness of the concept, and how they interpret it using a qualitative approach. Participants were recruited using a snowballing, purposive sample approach, resulting in 25 sheep farmers and 11 industry actors (veterinarians, farming organizations, advisors, and supply chain) being interviewed. To collect data, a combined approach involving semi-structured interviews and a facilitated workshop were used between April 2021 and March 2022. Data were then thematically analyzed using a hybrid of inductive and deductive coding process. The findings suggested that the perceptions of farmers and industry actors in the study regarding positive welfare differ from contemporary academic discourses. Overall, around 7 of the farmers equated positive welfare with “positive stockmanship”, while six of them expressed “good animal welfare” definitions associated with the Five Freedoms. In contrast, most industry actors (6) expressed interpretations associated with high welfare standards (going above minimum recommended practices) and positive mental experiences (3). Emerging discourses revealed the link between self-identity, social identity and what positive welfare is, the importance of knowledge exchange, and the need for practical indicators through language rephrasing. There is a clear need to enhance and improve knowledge dissemination strategies, particularly in the UK, where much research is being conducted on positive animal welfare.

## 1. Introduction

Animal welfare science continues to evolve as new research expands our understanding of the best ways to meet the needs and desires of farm animals to ensure they have a good quality of life. Initially, scientists proposed single-attribute, value-based definitions, including subjective experiences, adaptation strategies, coping mechanisms, biological functioning, harmony, natural living, and suitable environments and care [1,2]. These definitions did not reflect the complex nature of animal welfare, including its ethical dimensions, and were, therefore, condensed into a comprehensive concept built upon the interaction of subjective experiences, biological health and body functions, and expression of natural behaviors [3,4]. Subsequently, scientific research focused on these areas, especially subjective experiences, and biological functioning, while the expression of natural behaviors is also considered important within scientific understanding. The focus on subjective experiences, mostly negative experiences, such as pain and suffering, has now shifted toward consideration for positive animal experiences, such as pleasure and satisfaction, as part of determining an animal’s quality of life. This “new approach” is termed “positive animal welfare”.

According to analysis of available reports, positive animal welfare development can be grouped into three eras: pre-2010, 2010–2018, and post-2018 [5,6,7]. The pre-2010 era saw the concept slowly gaining scientific attention following works on animals’ affective states by Fraser and Duncan [8]. Later, animal physiologists and etiologists from Europe made important contributions by reviewing appropriate assessment indicators, thereby strengthening the concept [9,10]. In the United Kingdom (UK), the concept was brought into the scientific “limelight” with the Farm Animal Welfare Committee’s report advocating for a good life for animals, as the highest standard of quality of life for the animals [11]. After this, from 2010 to 2018, research on the subject focused principally on framework development by a number of research scientists to define benefits of positive welfare to farm animals based on the available animal welfare literature. Notable are the resource tiers framework [12] and Five Domain Model [13] developed in the UK and New Zealand, respectively. The resource tiers framework, known as a good life framework, is composed of increasing opportunities that provide positive welfare experiences to animals, while the Five Domain Model encompasses the internal physical/functional and external/social aspects related to welfare, with their consequences feeding into the affective domain. Here, the impact of positive and negative mental experiences can be used to determine the animal’s welfare state. Despite fifteen years of development, the good life framework has seen limited application on farm. However, the concept has been adopted by organizations championing higher welfare in the UK [14,15].

After 2018, positive animal welfare expanded into four interlinking themes: positive affective states, positive emotions, happiness, and quality of life [16]. These were subsequently distilled into hedonic positive welfare, which is built upon the animals’ likes and wants, with the aim of maximizing pleasure, and positive welfare balance, which aims to maximize positive experiences while also minimizing the negative ones [17]. Rault et al. [17] opined that positive welfare can be a distinct subject within animal welfare science if considered from the hedonic positive welfare point of view. However, recent biological advances have challenged this reductionist view by arguing that more “positives” can be experienced beyond hedonism when eudaimonia is considered [18]. The idea of eudaimonism (the state of being in “good spirit” across a lifetime) evaluates an animal’s quality of life at specific points over its lifespan [18,19].

This research aligns with the position that positive welfare is not necessarily a new overall concept. Therefore, it views positive welfare as encompassing subjective language to describe the *optimal* states of animals, characterized by positive behaviors (such as social interactions), ideal physiological functioning (such as resilience and thriving), and positive experiences, encompassing both affects and emotions (such as pleasure and satisfaction). This definition is not new per se, as it is grounded in animal welfare literature. However, it offers an integrated, measurable perspective that addresses limitations associated with previous definitions.

Current trends in the literature focus on several key areas, as summarized in [20]. These areas include indicators and assessments [21,22,23,24], eudaimonia and positive well-being [18], positive experiences as mitigators for negative experiences for net welfare states [25], global implications of positive welfare [6], comparing and bridging scientific and societal perspectives [26], views, behaviors, and attitudinal responses [27], as well as participatory engagements [28]. Its growing importance has also led to grants and funding being awarded to researchers to continue to make meaningful contributions toward good life provisions for animals [20]. While the aforementioned studies show an upward trajectory for positive animal welfare, they also indicate that positive animal welfare research has remained confined within small academic circles. The implication is that existing biases and limitations, such as viewing positive animal welfare from a reductionist point of view only, could perpetuate misunderstandings and misconceptions around the concept, thereby creating knowledge gaps between scientists and the agricultural and veterinary communities. Considering this, it is important to critically evaluate how to make current scientific advances more inclusive and representative of key stakeholders and end users of positive welfare.

This research aims to answer the following research question: how well is positive welfare known among UK sheep farmers and industry actors, and what are the implications for knowledge exchange and practice? The objective is to raise awareness, interpretation, and problematization of the concept among UK farmers and industry actors and see whether any new concepts may emerge. This study will provide real-world understanding with the aim to bridge the gap between theoretical concepts and existing and future practical applications of positive animal welfare. The findings can inform future research directions, policy development, and address the critical need for improved knowledge dissemination and participatory, co-design between academic circles and farming communities, as the concept transitions from theoretical aspirations to become an integral part of on farm practice.

## 2. Materials and Methods

This qualitative study adheres to the exploratory-descriptive method proposed by [29]. This approach, deeply rooted in sociology, is suitable for topics with scant literature and unresolved practical applications, enabling participants to contribute new insights to the field [29]. Natural scientists often question such qualitative methods employed by sociologists, perceiving them as subjectively developed [30,31]. Instead, they tend to expect sociologists to adopt a positivist methodology, with the belief that it ensures quantifiable data and objective analysis [31]. However, prioritizing statistical analysis in research can limit the scope of the research, and overlook contextual complexities that require in-depth exploration and explanation [30,31]. For example, Vigors’ [32] psychoanalytical approach to positive welfare focused on mental abstract conceptualizations, which does not focus on the importance of social and cultural processes in forming attitudes and opinions or the role of lived experience plays in influencing perceptions. However, social and cultural aspects have the power to uncover in-depth meanings and interpretations which can reveal important hidden opinions on the topic. Such emergent findings set the groundwork for future more positivist, quantitative studies.

### 2.1. Sampling

This study employed a snowballing and purposive sampling strategy in line with sociological recommendations [29], to capture various roles and regions within the UK sheep farming industry. This enabled us to construct a nuanced understanding of how positive welfare is conceptualized, interpreted, and understood across multiple contexts. The convenience snowballing technique was used to recruit participants. Farmers involved in positive welfare and wool projects across the UK were initially invited to join this study. The initial participants were then asked to recommend more farmers, leading to a snowball effect. Social media platforms (X, formerly Twitter, and Facebook) were used to broaden the reach. Industry actors were recruited through referrals, leveraging the professional networks of the researchers to identify key stakeholders across the sheep industry. The final sample comprised 25 sheep farmers and 11 industry actors. For the farmers, a purposive sampling approach was adopted to ensure representation from Scotland, England, and Wales to capture regional variations, if any. The eleven industry actors represented a cross-section of the sheep sector, including two veterinarians, one supply chain certificate agents, three agricultural advisors (two specialized in sheep and one general), and one academic researcher. Four other supply chain actors, including three from the wool sector and one from the meat sector, were also involved to offer views from the value chain. Sociologists justify sample size of about thirty respondents as an appropriate size that potentially allows themes and subthemes to emerge from the data [29], thereby generating new knowledge. Our sample size of thirty-six therefore falls within acceptable threshold in sociological methodologies. In addition, we also adhered to the principle of data saturation to enhance rigor in determining the sample size. The data saturation principle is widely used in determining sample size, in that it allows data collection to be continued until thematic repetition occurs in the dataset without the emergence of any new information [33].

Qualitative methods used to collect data were through individual semi-structured interviews and a facilitated group workshop [29]. Semi-structured interviews were conducted between April 2021 and March 2022, while the facilitated virtual workshop was held in March 2022. How these methods were adopted and applied in this study are explained in the following sections.

### 2.2. Semi-Structured Interviews

Semi-structured interview questions were designed following a “reflexive and iterative” process [34]. This means that interview questions continually evolved following an iterative literature review of existing knowledge as well as the authors’ reflection on the research journey. Therefore, the interview questions were “messy” during the research inquiry phase and were not structured until the final set of questions was determined. This “messiness” is an advantage, as explained by [34], because it helps researchers clarify the purpose of the research and connect them to the subject being studied. As an example, one of the initially drafted questions aimed to explore contemporary animal welfare and included the concepts of “lead” and “lag” indictors. However, after the interview questions were tested in the pilot stage, the drafted “lead” and “lag indicators questions were removed as they were mainly related to “negative welfare” attributes, which did not align with the positive welfare narrative developed in this research. The final interview questions included open-ended questions on the interpretation and understanding of positive animal welfare, its indicators, (potential) applications, perceived usefulness and benefits, and its knowledge dissemination. Interviews often commenced with queries about the interviewee’s farm background. Participants were then asked specific questions, following predetermined themes for the final draft of the questions.

Two tailored questionnaires were developed: one for farmers, with focus on animal welfare practice and included demographic information, and another for industry experts, focusing on knowledge dissemination and animal welfare corporate communications. To present a comprehensive analysis, we compared the sheep farmers, primarily meat farmers (with wool as a coproduct), to other groups of actors, focusing on their broad contrasts and similarities. Adopting such an “overall picture” approach is appropriate for reporting broader themes and patterns emerging from the data [35].

Initially, the interviews were conducted in person on the farms, and subsequently adapted to audio/video formats due to social/physical distancing caused by COVID-19 [36]. Audio/video interviews were completed either on the Zoom video-conferencing platform or using mobile phones, with audio recordings only [36]. Farm and Zoom interviews lasted about 40 min, while the duration of the mobile phone interviews varied from 20 min to 30 min. Table A1 in Appendix A provides details of the interviews with the 25 farmers and 11 industry participants. The first three interviews were part of both the pilot phase and were conducted to help the researcher gain familiarity with interviews and assess the effectiveness of the questionnaire questions, as mentioned earlier.

### 2.3. Facilitated Workshop

The facilitated workshop explored further salient points identified during the interview process in detail. The other purpose of the workshop was to ‘sense check’ the initial findings of the study with the interview participants and to build and expand on the richness of these initial individual findings. It was conducted over Zoom audio–visual technology in March 2022, and lasted for one and a half hours. The design was based on [37]’s method and was rigorously enriched following [38]. The primary researcher prepared the workshop by setting goals, creating agenda, and managing participants through emails and other engaging methods. Potential knowledge inequalities (which was clear from the interviews), and engagement barriers such as technological inexperience or unfamiliarity, all of which are sources of imbalanced discussions were acknowledged [39]. Creative, immersive incentives such as visuals (photography) and video blogs, commonly incorporated in qualitative methods to address similar challenges, were adopted [39,40]. Video blogs in focus groups have been shown in the past to facilitate cognitive transformation among participants and encouraged real-time collaborative construction of ideas and perspectives [40]. The video blog in this study was of one of the farmers (identity not included here) providing an account of what their interpretation of “positive welfare” is. This was played at the start of the workshop which afforded the attending participants a “level playing ground” to form confident opinions on the topic while engaging in reflective thinking during the workshop, and was therefore an ideal creative choice in enhancing the richness of the participants’ discussions.

Twenty-four of the interviewees attended this workshop. While six moderators facilitated the workshop: primary researcher (M.M), two co-authors (J.E.S and L.M), two other experienced facilitators associated with the research (L.M and D.C.J.M). The third co-author (I.Y.H) attended as an observer. The sixth moderator (R.A.T) was the “meta-moderator” who managed the workshop’s schedule, logistics i.e., moving people into their breakout rooms (focus groups). The attendees were divided into five breakout rooms, with the first five moderators facilitating each focus group simply as neutral learning guides (Table A2 in Appendix A). Before the workshop, facilitators attended a training session, where their roles and other responsibilities were comprehensively addressed. Having this many facilitators, and with each knowing their roles, eased operationalizing the workshop, as also demonstrated by [37].

Each facilitator presented the prepared questions derived from semi-structured interviews in the focus groups through Microsoft PowerPoint slides. The questions were as follows: How would you define the term positive animal welfare? Do you use the term “positive welfare” to describe the well-being of your sheep? If not, what other terms do you use? Does the way we frame welfare language (positive vs. negative) influence how farmers implement welfare improvement on farms? What welfare language should we use to establish a common understanding between farmers and society? How can positive welfare be used to create more value for farmers? Each question was allotted a time limit of 10 min and, therefore, 50 min was allowed for 5 questions. This timeslot was considered sufficient, as the average focus group had five participants, so each participant had an average of two minutes per question. The facilitators’ role was simply to guide their breakout rooms toward their discussion by reading out the presented questions. The timing ensured each group had the same experience and minimized the effects of extraneous factors. Participants were also asked to volunteer as scribers as a further measure of inclusion. All these steps reduced the subjectivity bias of the co-authors, thereby increasing the internal validity of the workshop. At the end of the focus groups, there was a plenary session aimed at comparing and examining the thoughts raised across separate groups. Due to time constraints, this was led via summaries of what each group had discussed. There were no logistical issues within the workshop except for one participant joining by audio only.

Both semi-structured interviews and the workshop were recorded, and were transcribed verbatim using Otter.ai (Version 3.5.0-121bc514, Los Altos, CA, USA), providing an integrated speech interface to text transcription and translation application using artificial intelligence and machine learning. The transcripts were then imported into NVivo for Windows (Version 12 Plus, QSR International Pty Ltd., Victoria, Australia) for thematic discourse analysis.

### 2.4. Thematic Discourse Analysis

Thematic analysis is a qualitative technique that involves identifying themes in a recurring predictable pattern. It is commonly applied in qualitative research because of its epistemological flexibility, allowing it to be adapted for exploratory-descriptive qualitative approach [29]. This research adopted a systematic thematic analysis proposed by [41], to allow themes to emerge from the data. This systematic approach was chosen not only because it aligns with the objectives of finding new insights on the studied topic but also because of its rigor in code identification, which reduces subjectivity and improves the repeatability of the process [41].

### 2.5. Coding and Themes Generation

The coding process followed the first five systematic thematic analysis steps proposed by [41]. These are transcription, familiarization with the data and selection of quotes, selection of keywords, coding, theme development and conceptualization through interpretation of keywords, codes and themes. The five stages were subjected to three progressive methods of coding. The first stage aimed to find out “what the current understanding of positive welfare” was among the participants. Deductive and inductive hybrid coding was employed to see (i) what the interpretation of positive welfare was and, (ii) comparisons to see whether selected quotes, keywords, codes and themes match existing interpretations found in the literature [32]. The second progressive coding, mainly based on inductive coding, aimed to observe how the positive animal welfare concept was problematized by the participants. Following this, the final progressive coding aimed to see what new meanings were associated with the positive animal welfare concept. Each progressive step was rigorously reviewed, refined and vetted during team meetings with the co-authors.

### 2.6. Positionality

The lead author took a reflexive approach and remained conscious of their positionality throughout the research. The analysis involved an iterative process of data coding and reference to the literature to ensure relevance. The lead author also considered how their identity might affect data collection and interpretation, noting the participants’ questions about their research. Potential social desirability bias was mitigated by not revealing the iterative nature of the research beforehand.

### 2.7. Study Limitations

This study’s non-probability sampling and the small sample size limited the generalizability of the findings. The nature of this research was not to test a particular theory but rather an exploratory one that seeks to shed insights on an evolving topic in animal welfare science. In future research, random probability sampling could be adopted to increase the external validity of such a study.

Secondly, there was a low number of veterinarians, especially academic veterinarians, who are likely to have a better grasp of the concept than their farmer-facing counterparts. Future studies could engage academic and farmer-facing veterinarians for a broader perspective on their roles in disseminating knowledge on animal welfare.

Additionally, in-person interviews proved to be the most effective form of data collection, compared to mobile phone interviews. The latter recordings were susceptible to environmental factors, such as network issues and clarity problems. Additional note taking was used to compensate for any potential loss of clarity.

## 3. Results

The results are presented here by theme. The themes were awareness of positive welfare, acceptance of established meanings of positive welfare, differences in participants’ articulated meanings of positive welfare, problematization of the current terminology and concept of positive welfare, and emerging meanings of positive welfare.

### 3.1. Participants’ Awareness of Positive Animal Welfare and Barriers to Knowledge Dissemination

The semi-structured interviews began differently for the participants. For those who were not part of the positive welfare wool project (see Section 2.2), the interview questions started with “have you heard of positive animal welfare?” This open-ended question aimed at their awareness, which covers prior association with attitude toward, and knowledge of or action related to a concept. Following this “settling in” question, participants were asked in the interviews and the facilitated discussion workshop to identify alternative terms for describing the “welfare” of their sheep. This follow-up questions aimed to allow participants to “introduce” language they were comfortable with and remove any potential barrier that could hinder them from expressing their understanding of “positive welfare”. Figure 1 shows the results of this follow-up question on the preferred language used by participants describe “welfare” of their farm animals.

Farmers chose terminologies that reflect their farming practices and resonate with their personal values. Those prioritizing health and productivity mostly use “happy” and “healthy” in combination with one another to describe animals with a “good life”. Farmer 5, however, strongly criticized and refused to be associated with the term “happy”, arguing it to be unscientific, unprofessional, and overused, often linked to “smallholder” farmers. She favored the term “contentment”, which she described as an animal being unafraid, curious, well-fed, and comfortable. Her language choice reflects her desire to create a distinct identity and articulate a more precise understanding of positive animal welfare. Conversely, farmers who participated in assurance schemes, which require adherence to specific standards, mainly preferred “high welfare” as a term, because it was perceived as having a market orientation, thereby providing competitive edge over non-scheme farmers. “High welfare” also featured prominently in the farmer focus groups, where it was perceived as not only more “quantifiable” than “positive welfare”, but also as an approach that considers the animals’ point of view when making management decisions:

“High welfare is better because positive is just a word. It is just chucked around, is it not? If you say high welfare standards, then you really can measure that. You cannot measure positive. Positive is just a word. Positive is just an idea. If you have something you can measure, then it is easier”.(Farmer 20_grp)

“So, everything we do throughout the year is just trying to get as high welfare standards for those animals on the farm as possible, whether it is my sheep dogs, or anything else that is in the farm”.(Farmer 4_grp)

Among the industry actors, advisors, veterinarians, and certification scheme professionals all appeared to throw their weight behind the “high welfare” term. However, the wool supply chain participants offered a contrasting perspective on terminology. During both the semi-structured interviews and the discussion workshop, the wool supply chain participants expressed their support for the concept of positive animal well-being. They claimed that this term represents a positive experience that animals can have and is related to their specific circumstances. However, their priority shift from promoting positive experiences for sheep (which is the aspiration for positive welfare) to focusing on wool value addition suggests that wool actors were interested in shaping public perception through strategic language use. In any case, this issue of switching descriptive wording from welfare to well-being has also gained attention, with recent understanding reasoning that both positive animal welfare and positive well-being are unique and have their own separate literature and, therefore, separate discourse.

Participants’ awareness of positive welfare was coded to three levels. Low awareness referred to those who had not heard of or associated with the term before, or who were still struggling to understand it even if they had encountered it. Medium awareness comprised those who had heard of or associated with the term before and recognized it as part of animal welfare (knowledge) and those who had expressed their attitudes toward the concept but were not able to explain the rationale behind the attitudes. High awareness consisted of those who had heard or associated with the term before, could satisfactorily explain what it means, and were actively implementing, promoting, or disseminating knowledge on it. Table 1 presents the frequency of responses among the respondents.

Table 1 shows that 57 per cent (n = 20/35) of the participants, more than half, had low awareness of the concept of “positive welfare”. These included sixteen farmers who were not part of the previous positive welfare and wool project (Section 2.2), two farmers who were part of the previous project and two others industrial actors. One of the two actors with lower awareness worked with assurance certification schemes, potentially limiting their exposure to the positive welfare concept, as such organizations typically focus on meeting basic welfare needs or market-driven welfare outcomes. The other, a farmer-focused veterinarian, stated that they had not heard of “positive welfare”, and that the Five Freedoms framework remained their primary guide for delivering good welfare in practice. Among the twelve participants with medium awareness were seven industry actors (four of which were not part of the previous wool project) and five farmers, all of whom were part of the prior positive welfare wool group. Of those with higher awareness, two were industrial actors and one was a farmer, who was not only aware but actively involved in adopting positive welfare practices.

Farmers’ lack of awareness was found to be associated with inadequate knowledge transmission within their social networks, including veterinarians and the broader farming community. Table A3 in Appendix B illustrates this dynamic, showing that 20 (of the 25) farmers, including those previously informed, replied that positive welfare knowledge is not being circulated, and current discourses primarily focused on health issues and productivity [Farmers 1,6,7,11,12,13,15,16,19,21,22,23] or general sheep information [Farmers 2,4,8,9,10,17,20]. Interestingly, even farmers with medium or high awareness often refrained from sharing it with other farmers, arguing that their peers are not open to discussions about improving welfare practices due to attitudinal differences, identity-related factors, and behavioral disparities:

“I do find it very difficult [sharing my positive welfare knowledge with other farmers] because my experience and my attitude is very different from theirs. I find it quite uncomfortable being with them because they just do not understand where I am coming from, and I do not really feel inclined to talk about it very much. The things I have been telling you I would not tell them because most of them do not understand what I am talking about. Because they see me as such a maverick because of what I do [regarding positive welfare] with Merinos, we are already regarded as stupid.”(Farmer 5)

Industrial actors’ approaches to positive welfare knowledge varied across the supply chain. The wool sector tended to frame positive welfare knowledge within their business values, indicating a shift in mindset toward improving animal welfare and deriving value for farmers. In contrast, the meat supply chain identified a lack of appropriate terminology, while veterinarians and certification schemes emphasized that their dissemination and practices still center on the Five Freedoms. Two well-informed advisors indicated that knowledge about positive welfare is currently internalized within their organization and not openly discussed. They attributed this to a lack of resources for implementing a comprehensive knowledge dissemination program, though they expressed hope for future initiatives contingent on increased funding:

“From a stakeholder perspective within the industry, there is looking at animal health and welfare… also, the Five Freedoms… to be honest, the Five Freedoms are the main term regarding within the [anonymized]”.(Certification scheme 1)

“[At] the organization level, I suspect it [positive welfare] is missing to some degree… The positive welfare aspects are certainly new, and are probably [evolving]. I am not [sure] they are within any of our communication material. But it is evolving. So, I expect it will become an important part of how our organization considers animal welfare”.(Actor 2)

“Certainly, in terms of welfare outcomes [framework for farmers], we have not yet brought in a good life framework. That is our next target area. Okay, but we do think it is important, but we need to gain, get the resources together to allow us to do that”.(Actor 3)

In conclusion, the findings suggested that the awareness of positive welfare as a concept was largely low among most farmers, but industry actors showed a more varied level of awareness. This disparity in awareness seems to be related to language framing, especially for farmers who suggest that their “farming language” was deeply tied to their husbandry practices and personal values. Language, therefore, appears to be a principal barrier limiting positive welfare knowledge flow among and between farmers. Furthermore, at the industry level, actors who are aware of positive welfare are still at the stage of internalizing this knowledge rather than actively disseminating it. Funding resources was seen as a key enabling factor to improve knowledge dissemination on positive welfare.

### 3.2. Acceptance of Established Meanings Associated with Positive Animal Welfare

Table A4 in Appendix B summarizes respondents’ views and discourses on positive welfare during the interviews and the virtual discussion group. The number n in the table is higher than the total number of participants, as some participants gave varying descriptions in their interviews and in the workshop. The virtual workshop proved an effective platform that invited in-depth discussions and allowed for greater exploration of the issues explored, which may have been limited during the interviews. It also showed that the participants’ understanding and knowledge of the concept increased within the research lifespan. By categorizing the responses, it also allowed for a better interpretation of the various perspectives shared.

Table A4 in Appendix B presents the frequency of participants’ views and discussions from both interviews and the workshop. Participants who maintained their interpretations in both settings were counted only once to avoid inflating the response numbers. The key finding shown in the table was that the most common definition of positive welfare among the farmers was “positive stockmanship” (n = 7) (detailed in Appendix B). Burton et al. [30] defined this as being built of three values and behaviors, including proactive care, rather than reactive, as well as intuition and empathy toward farm animals. The responses from the farmers (detailed in Table A4 in Appendix B) showed strong alignment with this definition, with specific emphasis on proactiveness of care as well as intuition. The prevalence of this definition among farmers (n = 7), all of whom had no prior awareness of positive welfare, indicated the power among the farming culture practiced by these farmers. It also seemed to illustrate the value they placed on their intuition and tacit knowledge, whilst providing what they perceive to be good life experiences for sheep.

On the other hand, the least-understood interpretation was that of the actual positive welfare framework itself. Only two farmers (Farmers 5 and 10) with prior awareness clearly interpreted positive welfare as defined within the research literature. Two advisors and a researcher with an academic background provided clear interpretations at the industry level, focusing on enhancing the animals’ overall positive experiences.

The findings stressed the need for further open dialogue, cooperation, exchange of knowledge, and research to establish a common understanding and farmer ownership over positive welfare as a concept and lived experience for sheep.

### 3.3. How Participants Problematize the ‘Positive Welfare’

#### 3.3.1. Is It Even Needed?

In both the semi-structured interviews and the facilitated workshop, some farmers expressed skepticism toward positive welfare, questioning its necessity given that animal welfare is already a comprehensive concept (Farmers 4, 7, 10, and 20). These farmers suggested that positive welfare is just another industry buzzword, similar to ‘regenerative agriculture’, and struggled to distinguish it from negative welfare (Farmers 8, 10, 12, and 20). Some farmers became confused by practical implementation issues, viewing it as an academic term disconnected from the farming reality (Farmers 8 and 11):

“I have an issue with positive and negative because the definition of welfare is health, happiness, and well-being. So, I think welfare sums up what we should be doing rather than [using terms such as] positive and negative”.(Farmer 4)

“I am struggling with the concept, and I have asked the other guys on [xxx], and one of my friends she does not farm, and she really gets it. And the ones that farm, like I do, we are really struggling with it. So, is there not… a defined definition for how it works in practice?”(Farmer 8)

Industry actors offered additional insights. Actors 1 and 3 believed that farmers struggle with positive welfare due to greater familiarity with terms that focus on negative welfare aspects, such as pain and suffering. Actor 3 also argued that, without proper scientific guidelines, the concept invites misrepresentation, stating:

“I think the positive terms are not well understood, and I think that they are much more nebulous than the negative ones. I think the avoidance of pain is [understood] and avoidance of stress is better understood than the positive actions… Anybody could interpret it [positive welfare] anyway they like… I think it is a very high-level term rather than one [that can be readily applied] that so it is like anything”.(Actor 3)

“I think our farmers respond more to the use of the word health. They respond more to health rather than welfare; welfare has ended up being more connected to the conditions in which animals are kept in the more intensive livestock sectors I think”.(Actor 1)

In summary, farmers’ negative attitudes and concerns about the ambiguity and lack of clarity surrounding positive welfare and the problematization of the concept highlighted issues in knowledge dissemination, revealing a gap between academic concepts and real-world understanding of applications. Industry actors’ observations about farmers’ inclination toward minimizing pain rather than promoting positive experiences aligned with established psychological principles that humans often find it easier to conceptualize the absence of negative experiences.

#### 3.3.2. Issues with the Assessment of Positive Welfare

Farmer 8 strongly objected to the use of positive, anthropomorphic language as an assessment index for positive welfare. She contended that it is anthropomorphic and unsuitable as a business indicator. She emphasized the importance of measurability in business tools, and proposed thriving as an appropriate indicator:

“The question is are they [animals] thriving? [because] I think, well-being is very much a human term. And we can communicate well-being [but] I am not sure animals can communicate that on the same level. So, for me as a farmer, it is about, are they thriving?”(Farmer 8)

Farmer 1 introduced another dimension to this debate and argued that the ability to assess positive welfare indicators is deeply connected with the tacit knowledge of the farmer, making it an intrinsic part of a farmer. He claimed that imposing any formal measurement requirements for positive welfare would undermine the cultural tradition of transferring this indigenous knowledge to future generations. Farmer 14 also shared similar perspectives in both interviews and the workshop:

“That measurement [of positive welfare indicator] is in me. Is it important for me to pass that on to a layperson, so they can measure it? The answer is no. But it is important to pass it on to a new generation of the shepherd[s], and I am sure they would grasp it very quickly”.(Farmer 1)

Three industry actors suggested an alternative approach to address the challenges of positive welfare assessment. Acknowledging that the subjective nature of the concept makes practical assessment difficult, they recommended development of further positive welfare indicators through rephrasing objective, outcome-based indicators in a positive manner, as negative language could adversely affect farmers:

“If you think of outcome measures, you tend to record the negatives, if you record the number of lame or the number of whatever you would record the negative rather than see it in a positive light. And [it is better you record it as] I [have] got 93% sound sheep, [rather than] you would say 7% lame”.(Veterinarian 2_grp)

“And I think it is a nice idea, though, to really think about it in a more positive frame, framing rather than always looking at lack of bad things”.(Actor 3_grp)

In summary, some of the farmers appeared to be concerned about the practicality and measurability of positive welfare indicators, arguing that the indicators are anthropomorphic in nature and, therefore, removed from farming realities. In contrast, industry actors acknowledged the challenges but suggested rephrasing negative indicators into positive terms for clearer measurement.

#### 3.3.3. Dominance of “Health” and Its Inseparability with Welfare of Animals, Affecting the Definition of Positive Welfare Space

The sheep industry actors took a different approach in expressing their concerns with positive welfare. Almost all of them showed a positive attitude toward the term, arguing that the concept captures subjective, anthropomorphic concepts previously rejected in scientific assessments of positive welfare. Sheep advisors, as well as the two farmer-facing veterinarians (in two separate breakout rooms) and wool supply chain actors, agreed that there was a need for positive welfare. Researcher 1 provided justification for positive welfare, as follows:

“I think we are [becoming] more comfortable using, the words like happy and relaxed, pleasure, joyous, or something like that—words we are happier with using. But I suppose there has been this fear of introducing non-scientific and anthropomorphic terms because we have all been slammed over the past [for using those terms]. So, using positive welfare is the catch-all for something that sounds perhaps more scientific, but less subjective”.(Researcher1_grp)

However, Actor 1 highlighted that the sheep industry is dominated by health discourses, which the industry conflates with “welfare”, as they perceive them to be inseparable:

“Within our industry, I think we often feel that by improving health, we are going to improve the welfare of animals. So, the more we can improve disease control, parasite control, improved nutrition, the better that will be for the welfare of the animals. So, we often we regularly very regularly connect the two in that in that respect”.(Actor 1)

With the industry’s propensity to intensify animal health discourse, Actor 2 suggested that defining the “new” concept of welfare, positive welfare, becomes increasingly challenging due to health being the primary focus of the industry:

“So, with sheep, the language of welfare is so tied up within health so much that it’s sometimes, it is quite difficult to think even to think about it all, the positive, to the positive choices. And that we just tend to be quite entrenched in our views of thinking about health and improving health, and that being so associated with welfare. The positives can be quite difficult to think about”.(Actor 3_grp)

The takeaway from these discourses is that the sheep industry continues to focus on health issues, which influences how good farm animal welfare is perceived. This health-focused view of “welfare”, shaped by its perceived importance to farmers, impacts the industry dialogues on the meaning of welfare and, subsequently, positive welfare. To successfully extend the concept of positive welfare within the sheep industry, there is a need to first move beyond this health-dominated view of animal welfare. As a first step, this will require an increase in continuing the discourse on positive welfare with farmers through events, publications, and farm walks, as proposed by Actor 1.

### 3.4. Emerging Meanings in the Discourses of Positive Welfare

#### 3.4.1. Domestication and Institutionalization of Sheep

A theme generated through the facilitated discussion workshop was that positive welfare should not embody domestication and institutionalization, which were described as continually keeping sheep indoors and depriving them of novel experiences in natural environments:

“They are not dealing with novel experiences. They [the sheep] are dealing with the routine stuff only. Yeah. I mean, the routine might not be quite fun, but it is still routine”.(Actor 3_grp)

This quote highlights a belief that keeping sheep in domesticated and institutionalized conditions without exposure to novel experiences may negatively impact their welfare. The lack of novelty in their lives can lead to boredom and frustration, harming their mental and physical health. Furthermore, the lack of novel experiences can decrease their cognitive abilities, as they need the opportunity to learn new tasks or behaviors. It can decrease their quality of life, as they cannot engage in stimulating activities that could enrich their lives. Prior research on farm animals, while not specifically focusing on sheep, has indicated that institutionalization can lead to the development of negative stereotypic behaviors [42]. However, there is limited evidence to associate this with sheep. The same industry actor and another farmer strongly objected to promoting the widespread adoption of positive human–animal relationships, such as scratching the backs of sheep, brushing, and gently stroking, which elicit positive experiences in sheep and other livestock [43]. Such practices were seen as extreme expressions of affection and/or extreme anthropomorphism (propagation of human traits on to animals). Farmer 15_postgrp argued that extreme expressions of emotions toward sheep, such as gentle stroking, misrepresent the realities and practicalities of livestock farming. Farmer 15 expressed a cautious and practical opinion that such practices could be detrimental to farmers. This is because the portrayal of livestock animals as pets could cause a cognitive disconnect between consumers and the reality of livestock production. In other words, the farmer was concerned that if sheep are increasingly perceived as pets, this could lead to a change in public opinion and make consumers less willing to consume meat, causing economic harm to livestock farmers:

“I think you must be careful [in making consumers choose positive welfare] because if people think they are all pets and [have] got toys to play with, it might turn them off eating lamb chops. I am also concerned that if the public sees images in the film [video], it could turn them off eating lamb! No one would want to eat their dog”.(Farmer 15_grp)

In summary, farmers are concerned about the consequences of extreme farm animal anthropomorphism/affection. These concerns highlight a complex link between societal attitudes, expectations, and their potential impact on husbandry practices. While consumer concerns and expectations about the welfare of animals are important, it is equally important that ethical considerations are balanced with the reality of farming. Researchers, such as Stokes et al. [28], are already exploring ways of ensuring that a variety of positive welfare practices are available, which are ethical, practical, and achievable, but more studies are needed looking at the applicability and adoption of these practices in different systems, especially in the context of sheep.

#### 3.4.2. The Relationship between Self-Identities, Social Identities and Positive Animal Welfare

Another emerging theme from the study is the conflation between farmers’ self-identity, social identity and the concept of (positive) animal welfare. Self-identity in social sciences refers to a person’s distinctiveness in a group or role through their values, attitudes, and beliefs [44]. In the interview, Farmer 5’s discourse showed a strong connection between her self-identity and positive welfare practice. Farmer 5, a hypothetically early adopter, clearly expressed a different view of positive animal welfare to the rest of the farmers. She is implementing a specific positive welfare framework developed through academic consultation and reviewed quarterly with her veterinarian, reflecting her “individuality” rather than “similarity” with other sheep farmers [45]. She further highlighted her association to this “micro social group” involving farmers who are “practicing” positive animal welfare, which was unique and distinct from the wider social group:

“I think the typical average sheep farmer would think this [positive welfare] was a waste of time. It will not make any difference to the price he gets paid by the abattoir. So why the hell should he bother? There are very few of us that I know that do what I do. And sometimes, sometimes you must just keep quiet in a room full of sheep farmers… Because I am a woman. So, I am already an idiot. Because I’m female, I am an idiot. I do not know anything. I am an outsider. Because I was not born to the farm. I came from outside the industry. So, I am even more of an idiot. And I do not know what I am talking about”.(Farmer 5)

Farmer 3, who tends an endangered rare sheep breed, also strongly expressed conflation of self-identity with that of their animal welfare. They showed the distinctiveness of their “micro social group” by asserting they strictly adhere to agroecological principles. Farmer 3 further argued that their practices have given them a reputation for welfare. Considering their self-identity, Farmer 3, without prior awareness of positive welfare, likened it to industry standards and suggested that it is limiting in its scope:

“There is a slight reference to the Five Freedoms. We are going to go a long way beyond that. Because we do not just want animals [to be] comfortable animals; we want happy and joyful animals, that you can truly enjoying life and live it to the fullest, especially for those who only have two years of life. Yeah, we are way beyond that”.(Farmer 3)

Farmer 3’s quote has some interesting meanings. They described their own “view” of animal welfare as an animal that is happy, joyful, and living life to the fullest. By implication, this sentence can be taken as a principle of positive welfare because it is a part of the animal’s overall quality of life. This illustrates the issue of knowledge dissemination of the concept, as some practices of positive welfare are already being implemented by these farmers but without them knowing so.

Farmers 1, 8, and 14, on the other hand, saw themselves as “in-groups” of the traditional farming group, and therefore held similar views with contemporary farmers, all in contrast to the members of the outgroups [45]. These farmers perceived positive animal welfare and its promoters as members of the “outgroups” who do not see “farming” from the same perspectives as them. This perceived “in-groups” versus outgroups” dichotomy prompted frustration and resistance. In shifting from the “in-group” identity within the traditional farming groups, to the “out-groups” identity in positive welfare farming, these farmers raised concerns:

“I mean, if I went to the meeting this evening and sat around with farmers that are my age and older {and talked to them about positive welfare], I think they go, you have lost the plot now, you have been to university and you have really lost the plot.”(Farmer 8)

“I think a farmer will have a feel for his sheep, that they are happy. Because I know how the sheep react as soon as I go in the field. And that is something I could not explain to a non-sheep keeper, but most sheep farmers will know exactly what I am talking about. So that is quite difficult to explain to the layperson”.(Farmer 1)

“We as farmers, we know our animals, we know what is right, we have absorbed the ambiance of the sheep, which people from outside our industry, who are stakeholders in our industry, but not at grassroots with the livestock level can be critical to”.(Farmer 14)

#### 3.4.3. The Interconnectedness of Humans, Handling Systems and Genetics

Actor 2 in the industry actors’ focus groups anticipated challenges in defining and implementing positive welfare from a sheep farming perspective. The actor believed that there was already the assumptions that sheep inherently have “positive welfare” simply by living in the natural environment as opposed to animals confined in intensive systems. Other industry actors, including Actor 3, the farmer-facing veterinarian and even Supply Chain 4 from the wool sector, all echoed these sentiments and emphasized the need to “develop what is already there rather than change the wheel”. Therefore, in a free-flowing discussion, this group of industry actors explored a “new approach” to positive welfare by drawing on three interconnected factors: more human intervention, handling practices and genetics.

The farmer facing veterinarian proposed that positive welfare could be achieved in sheep with “more human intervention” such as early disease identification and treatment, and use of analgesia for mitigate pain during husbandry practice. Actor 3, whose secondary occupation was sheep farming, pointed out the challenges of more human intervention on sheep, especially as it entails “catching the sheep” (i.e how the sheep are handled). The actor insinuated that good sheep handling practices were related to several factors, including sheep farmers’ own welfare. Considering these concerns, Actor 2 suggested the need to increase focus on, and to promote breeding sheep for resilience and coping strategies, with further discussions held around the implications for such breeding on the behaviors of the sheep. In summary, a new contention for “positive welfare” in sheep farming was formulated and centered on practical realities of more human intervention, handling practices and genetics, considering what is achievable for the farmers in promoting good life experiences for their animals.

## 4. Discussion

This research aimed to explore United Kingdom (UK) sheep farmers’ and industry actor’s awareness and perception regarding positive animal welfare. Findings revealed that most farmers interviewed, or part of the focus group were unfamiliar with this concept, and they did not understand what it meant. Regarding the self-reported lack of knowledge dissemination by and between farmers, as well as with their veterinarians, social networks appeared to contribute to the lack of knowledge on the concept. Farmers instead expressed their preference for less anthropomorphic language, such as high welfare and good welfare, to describe what a good life means for the sheep. In essence, the self-reported lack of awareness and knowledge led to misunderstandings of what positive welfare is and could subsequently affect its adoption by farmers. On the other hand, industry actors had better knowledge and awareness of “positive welfare”. Despite their knowledge, their discourse on positive welfare has mostly remained internalized, or is influenced by a business focus. Health issues seemed to be the main discourse in the industry, severely impacting the dissemination of positive welfare knowledge to the wider farming community. The findings of this study have implications for positive welfare adoption, given that the UK, is not actively disseminating information on it. There are implications for the topic in this regard, as it continues to experience internalization of knowledge among a small group of academics. Therefore, there is clearly a need for further research and knowledge dissemination initiatives to explore optimal knowledge dissemination strategies from scientific to sheep farming communities, potential barriers, as well as the impact of such “new” ideas, such as positive welfare, on changes to industry focus.

Research on the impact of changes brought about by progressions in agricultural thinking and practices on farmers has recently revealed interesting insights. Hammersly et al. [46] highlighted the impact of bringing forward new policies to replace existing ideas that farmers are acquainted with, not only on their traditional views but also their identities and well-being, particularly among male farmers, who feel a loss of autonomy and expertise. Vigors et al. [47] further showed that farmers’ identification with traditional husbandry practices, despite the shift in scientific concepts to include positive mental experiences, remains unchanged. Our findings indicated that farmers may resist agricultural innovations due to two more reasons, their opposition to the use of anthropomorphic language, as well as their view of positive welfare and its promoters as “out-groups” that are not part of the “traditional farming” social groups. These two kinds of opposition have serious implications when promoting positive welfare practices. In the case of the former, it suggests that changing farmers’ behavior through framed language, for example, clearly engenders resistance to the concept, and could therefore lead to rejection and subsequently the non-adoption of “new” farming ideas. Therefore, although our study has not resolved this emerging paradox of opposition to anthropomorphism while promoting good life/positive experiences for sheep, one of the ways forward is to consider exploring positive welfare’s framing and its implications for farmers’ behavior and attitudes. For the latter, it suggests that farmers traditional group- and self-identities remains a key factor for engaging with positive welfare, given how identification is conflated with animal welfare. Therefore, there is a need to explore ways to align the attitudinal, behavioral and cognitive interaction of farmers traditional groups with the principles of positive welfare and, therefore, provide insights on how to engage farmers in and redefine positive welfare within their practical realities.

## 5. Conclusions

In conclusion, this study revealed interesting findings that warrant further scientific exploration and consideration when designing any knowledge exchange initiative around the subject. While industry actors had knowledge of the concept of positive welfare and viewed it with a positive attitude, UK sheep farmers perceived it as an external imposition that did not align with their group identity, leading to resistance. This resistance was further intensified by the theocratic opposition to anthropomorphic language in the positive welfare discourse. Despite this resistance, farmers demonstrated the ability to express their emotional connection to their animals using emotive-evoking, positive language. Scientists and other engaged actors need to engage with farmers’ discourses to improve their communication on positive welfare concepts through knowledge dissemination events to overcome these barriers. Furthermore, methods and strategies should be put in place to encourage existing farmers with an understanding and positive view of the concept to engage in dialogue and share information and practical examples with their peers. Moreover, future studies can explore the effect that welfare language framing has on resistance or acceptance to merging concepts. Doing so may allow further insight into other social and cultural factors which affect the engagement, receptivity and uptake of such concepts in practice.

## Figures and Tables

**Figure 1 vetsci-11-00452-f001:**
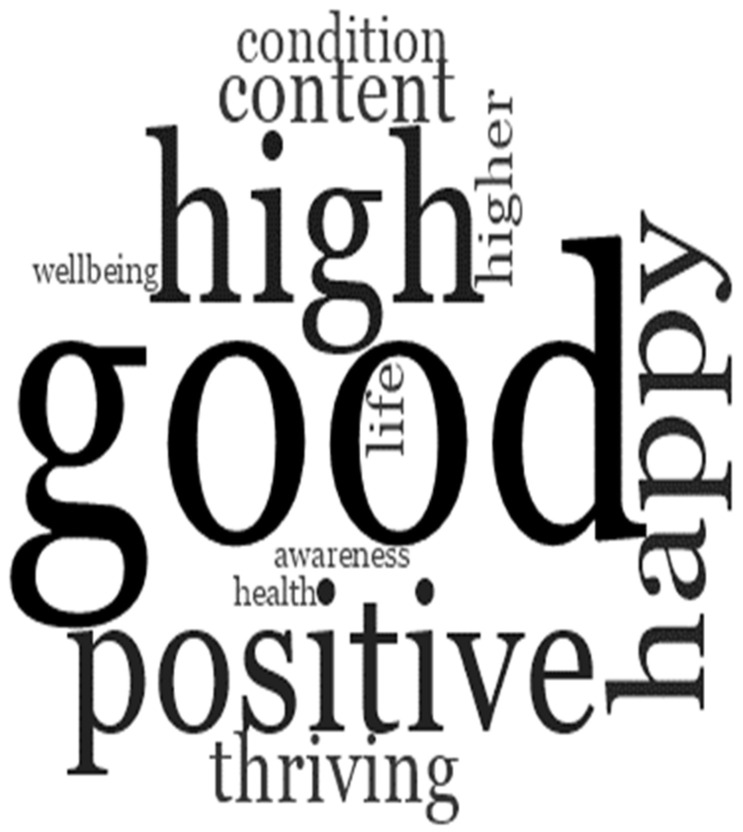
Word cloud displaying participants’ preferred terminologies for describing positive animal welfare.

**Table 1 vetsci-11-00452-t001:** Interview participants’ awareness of positive animal welfare (generated from NVivo).

Level of Awareness	Example Quotes	Frequency and Category of Respondents	Total
High awareness	“I think I understand the context—that is, welfare enables an animal to lead a good life rather than just avoid negative experiences”.	2 industry actors 1 farmer	3
Some awareness	“I think positive. (yeah) Animal welfare, yes. But no (not heard of positive welfare)”.	7 industry actors 5 farmers	12
Low awareness	“(Positive welfare) is not a term I heard of over here”.	2 industry actors 18 farmers	20

The frequency here captures the number of participants who specifically responded to the awareness of positive animal welfare question.

## Data Availability

Data are available upon request from the corresponding authors.

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
