# Peer review of "Discourses on Positive Animal Welfare by Sheep Farmers and Industry Actors: Implications for Science and Communication"

_vetsci, 2024, doi:10.3390/vetsci11100452_

Round 1
Reviewer 1 Report
Comments and Suggestions for Authors
vetsci-3145458
Perspectives of UK Sheep Farmers and Industry Actors on Positive Animal Welfare
I have several concerns that require consideration.
1. The paper is really in applied sociology, and it is submitted to a veterinary journal. This may be a conscious choice of the authors as the paper may get more exposure from the animal production-veterinary community that it would from the sociology academia.
2. Scientific papers are identified by hypothesis testing and open with a null hypothesis and describing the challenge to defeat the current theory. There is really no hypothesis to test in this paper. A near null hypothesises could be the statement “livestock producer’s and the associate production complex do not have a clue what the term “positive animal welfare” means”.
3. Positive animal welfare, the focus of the paper is not clearly defined in the paper. The near approximation is at line 515-517 The findings stress the need for further open dialogue, cooperation, exchange of 515 knowledge and research to establish a common understanding and farmer ownership 516 over positive welfare as a concept and lived experience for sheep.
4. The authors do not deal with the central paradox of the paper which is how do we reconcile our theocratic opposition to anthropomorphism yet promote an understanding of the lived experience for sheep. I have no real recommendation other than stating the logical dilemma.
5. The authors avoid discussing the ontology of animal well being and the emergence of “positive welfare” as a construct. The paper should indicate that positive welfare might be just the current fad in the endless cycle of researchers applying for grants and granting agencies justification of funding something new and progressive. The reality is that positive welfare is an academic pursuit not a real issue for the farming community.
This may be a concern to the editorial staff in the paragraph Starting The “good life framework” line 54-63 the authors cite references 7,8,9 & 10. Are from the journal Animals, Animals, Animals (#9 journal not named) and Animals which is a MDPI publication as is the journal veterinary sciences, the target for this review. There is a bullet in the instructions to reviewers
- Are the cited references mostly recent publications (within the last 5 years) and relevant? Are any relevant citations omitted? Does it include an excessive number of self-citations?
Can publishing houses manifest a self-citing bias? I am concerned that one could infer that the “good life framework” is a creation of MDPI and not a generally considered construct in the wider animal health and welfare discourse. A google search of “positive animal welfare” 11:22 2024-07-25 USA Central time, returned these first 10 references in this order:
1. What is so positive about positive animal welfare?—a critical review of the literature AB Lawrence, B Vigors, P Sandøe - Animals, 2019 - mdpi.com
2. Positive animal welfare states and reference standards for welfare assessment DJ Mellor - New Zealand veterinary journal, 2015 - Taylor & Francis
3. Positive animal welfare states and encouraging environment-focused and animal-to-animal interactive behaviours DJ Mellor - New Zealand veterinary journal, 2015 - Taylor & Francis
4. Behavioral diversity as a potential indicator of positive animal welfare LJ Miller, GA Vicino, J Sheftel, LK Lauderdale - Animals, 2020 - mdpi.com
5. Enhancing animal welfare by creating opportunities for positive affective engagement DJ Mellor - New Zealand veterinary journal, 2015 - Taylor & Francis
6. Citizens' and farmers' framing of 'positive animal welfare' and the implications for framing positive welfare in communication B Vigors - Animals, 2019 - mdpi.com
7. Assessment of positive welfare: A review JW Yeates, DCJ Main - The Veterinary Journal, 2008 – Elsevier
8. Animal emotions, behaviour and the promotion of positive welfare states DJ Mellor - New Zealand veterinary journal, 2012 - Taylor & Francis
9. Moving beyond the absence of pain and distress: focusing on positive animal welfare PV Turner - ILAR journal, 2019 - academic.oup.com
10. Extending the 'Five Domains' model for animal welfare assessment to incorporate positive welfare states DJ Mellor, NJ Beausoleil - Animal Welfare, 2015 - cambridge.org
There is a overrepresentation of the journals Animals and the NZVJ which may appear to be a bias. A review of the first author profile at https://www.researchgate.net/profile/Mukhtar-Muhammad lists 2 publications both in the MDPI journal Animals. I strongly recommend in the introduction to add a fairly thorough review of this construct and identify the birthplace of “positive animal welfare” and Mellor in New Zealand as the origin. This project was completed on residents of the UK and this should be pointed out in the conclusions the challenge of knowledge diffusion.
6. There is no section in the paper on “limitations of this study”. The most eye poking for me was that there was a single veterinarian identified when the British Veterinary Association has a whole division in the “Sheep Veterinary Society” which was not approached as a source of opinions. In the text the plural of veterinarian occurs primarily; eg. The lack of focus on health was problematic for veterinarians. One veterinarian sug- line 614 is misleading as there was only a single veterinarian in the whole study and using the plural is inaccurate. This obfuscation was consistent in: including veterinarians (line 12). Line 25-26, line 382, line 640.
7. A issue that was not dealt with was the conflation with farmer questionnaires where good animal welfare leaks into the self identification of a “good farmer” as in Vigors B, Wemelsfelder F, Lawrence AB. What symbolises a “good farmer” when it comes to farm animal welfare?. Journal of Rural Studies. 2023 Feb 1;98:159-70. (Vigors 2019 paper in Animals is cited, No. 30) may also consider Hammersley, C., Meredith, D., Richardson, N., Carroll, P. and McNamara, J., 2023. Mental health, societal expectations and changes to the governance of farming: Reshaping what it means to be a ‘man’ and ‘good farmer’ in rural Ireland. Sociologia Ruralis, 63, pp.57-81.
I think it appropriate to identify this issue (Sefl identity conflated with animal welfare) especially in reference to the quotations from Farmer 5 who had a massive identity investment in the raising of sheep. She would not do well in the western plains of North America where we lose 20% of lambs to carnivore predation, our primary welfare issue.
8. I did not understand the clause “design fiction” which appears 4 times between line 150 and 168. I really have no idea what this construct is. I downloaded and read the Steve North paper and do not feel that much mor enlightened. For me this concept is a n orphan thought which does not occur in the results or the discussion. If possible, I would prefer to have the Section 2.3 Facilitated workshop, rewritten with the words design fiction removed. I think it does not add to the paper and is a bit of a distraction.
9. I have not seen the word petrification in the discourse of animal welfare and that may be explained by my geographic location in Western Canada. You may want to identify this as a new meaning of the word as Springer has a book called Petrification Processes in Matter and Society, part of the book series Themes in Contemporary Archeology https://link.springer.com/book/10.1007/978-3-030-69388-6 .
10 the references need some checking
4-looks suspect no journal, also 9, 11, 30, 32, 36. Ref No. 28-no publisher of a book,
7.5. Rating the Manuscript
During the manuscript evaluation, please rate the following aspects:
- Novelty: Is the question original and well-defined? Do the results provide an advancement of the current knowledge?
- The advancement of knowledge is limited to moderate.
- Scope: Does the work fit the journal scope*?
- I have a weak opinion on this standard I do not read this journal with any regularity.
- Significance: Are the results interpreted appropriately? Are they significant? Are all conclusions justified and supported by the results? Are hypotheses carefully identified as such?
- This is a failure I delt with lack of null hypothesis above.
- Quality: Is the article written in an appropriate way? Are the data and analyses presented appropriately? Are the highest standards for presentation of the results used?
- Writing is pretty solid, however; it is pretty much a bedtime story without significant numerical statistical analysis as there was never an intent to go beyond narrative reporting. The authors do not find any real (statistical) significance of their arbitrary group in of participants.
- Scientific Soundness: Is the study correctly designed and technically sound? Are the analyses performed with the highest technical standards? Is the data robust enough to draw conclusions? Are the methods, tools, software, and reagents described with sufficient details to allow another researcher to reproduce the results? Is the raw data available and correct (where applicable)?
- I have read a lot of this type of social science investigation, and I find this one pretty par for the course. The methodology is completely compatible with current practice in social studies research.
- Interest to the Readers: Are the conclusions interesting for the readership of the journal? Will the paper attract a wide readership, or be of interest only to a limited number of people? (Please see the Aims and Scope of the journal.)
- With this journal the editors have pretty broad discretion - It publishes original research articles, reviews, communications, and short notes that are relevant to any field of veterinary sciences, including prevention, diagnosis and treatment of disease, disorder and injury in animals.
- I don’t think a lot of people will read it. There is a general expectation that as national veterinary infrastructures assure phytosanitary standards for international trade that certification of production systems for international trade will eventually shake out as a function of national veterinary service. Currently there is private competition for certification as a business opportunity. In North America the heart of neoliberal fanaticism we have PAACO Professional Animal Auditor Certification Organization https://animalauditor.org/. It is reasonable to publish news on the battle between various animal science professions for dominance in the field of commercialized animal welfare.
- Overall Merit: Is there an overall benefit to publishing this work? Does the work advance the current knowledge? Do the authors address an important long-standing question with smart experiments? Do the authors present a negative result of a valid scientific hypothesis?
- This is solid work, the bread and butter of academia in pursuit of tenure, a manifestation of postmodern neoliberal university management.
- English Level: Is the English language appropriate and understandable?
o The English is perfect, and I have no reason to suspect that it was not carried out in accordance with generally accepted ethical research standards.
Reviewer 2 Report
Comments and Suggestions for Authors
The study investigated the attitudes of UK sheep farmers towards positive animal welfare. The researchers conducted structured interviews and a virtual discussion-based workshop with sheep farmers and a selection of other stakeholders. The overall impression seems to be that the farmers would prefer more simplistic language and some value animal health over the animal’s positive experiences. The researchers recommend more communication between academia and the industry. Without an estimate of the number of UK sheep farmers, it is difficult to assess whether the sample size is appropriate to represent the UK sheep farmers in general. The methodology seems thoughtful, though without any anonymity, I would be worried about the reliability of the data. The results are presented unfiltered and unbiased with a great many quotes (maybe too many?). The conclusion appears supported by the data presented. A very interesting study with some funny, some heartwarming, and some eye-opening quotes. Here are my comments:
Abstract and summary
Please provide an example of an interpretation associated with ‘high welfare’?
M&M
L92-94: Could you elaborate on how the questionnaires for farmers and stakeholders were different?
L106-107: please provide an estimate of the number of sheep farmers in the UK that would have qualified for you study so the reader can get an idea of the external validity.
L153 ff: Can you elaborate on how the discussions were lead? Were there instructions for the different facilitators? Did they ask questions or simply transcribe and forward the slides?
Tables 1 and 2: please add a row with the total n at the bottom or top, so that the reader can easily get a feeling for representation of each response.
L287: Full stop after ‘thrive’.
L320: Is ‘happy health sheep’ a known term or concept? If not, please reword.
Table 2: The happy, healthy definition depends on your response to my last comment. It might require some rephrasing.
L393: Who is Advisor NA? I might have missed something, or it’s a typo of some sort.
Figure 1: This word cloud includes some terms that are not intuitively understood. The cloud might include too many terms or could be visualized differently. For example, how do the terms ‘research’ or ‘red’ apply here?
L779: Did you invent the term ‘petification’? It feels like a valenced term to me that makes minimal human-animal interactions into a problematic thing with its own term.
General remarks:
Could you please elaborate on the potential consequences of conducting interviews vs anonymous self-reports. What are the risks of the participants being untruthful or withholding information?
The sheer number of quotes is a little overwhelming to the reader. I would suggest re-evaluating which ones need to be in the main text and which could be moved to the appensix.
Reviewer 3 Report
Comments and Suggestions for Authors
Thank you for this very nice article, I have some questions and recommandations:
- Line 43: Not clear « positive experiences » please rephrase
- Line 45-46: … prove their overall wellbeing states through …
Is it that the meaning? Otherwise there are a repetition in this definition.
- Line 62: …”engagement at all stages of design …
Replace by: Stakeholders such as….
-Line 101-102: Is there a bias by including farmers who are already aware and have good knowledge and experience in this kind of study? This should be mentioned as a limit of the study too even if this helped to encourage other participants.
Line 163: why you choose “grazing environement” as “indicators” is it related to local practices in UK? If we do the same study in Europe for example, the concept and answers should be different?
Line 2010 and Table 1: How did you choose the level of awareness? Is there specific criteria?
Line 311-325: could you add the frequency of each “meaning”?
- How did you perform Figure 1? Website? Please add this onformation
-Line 816: perhaps we have to use the two words: “positive animal wellbeing and welfare” as the meaning is different?
Reviewer 4 Report
Comments and Suggestions for Authors
Comments to the Author
Comments:
Manuscript ID: 3145458, Title: Perspectives of UK Sheep Farmers and Industry Actors on Positive Animal Welfare
General comments
The article is about an important and interesting topic. However, it also has significant weaknesses. For example, there are difficulties with the authors' own definition of positive animal welfare, lack of originality; inferences are made based on very subjective or descriptive questions, without proper statistical analysis, as well as with other issues of design, analysis, interpretation and conclusions. Authors can review some comments below.
No result values are presented in the abstract.
Conclusions are not presented in the abstract.
Several keywords are repeated in the title, this should not happen, just check the meaning of the keywords in a scientific article.
L43-47: This is the authors' definition of positive animal welfare? Do not use circular phrases to define terms. What really is the authors' definition of "positive animal welfare"?
L64-82: Based on what is mentioned here, what is the novelty of this study? I do not see any real novelty or originality in the work, in this sense, the authors should justify with solid arguments the reason for this study.
In Materials and Methods and Results there are numerous questions, comments and doubts, I will only mention some of them.
Was this study approved by a human ethics committee?
L106: n = 25, Was a sample size calculation done for this study?
L111-114: On what basis? It is extremely diverse, for example only one veterinarian? Is this real?
L127-130: But in L98 it says it was in 2021-2022, check.
L166-167: Why do the authors mention figures that are not presented in this paper?
L202-203: Is it really necessary to clarify this?
L206: Can the authors really make this claim that the age of farmers affects their awareness of positive animal welfare..."?
Is the "n" correct?
What test did the authors use to arrive at an inference of that degree?
How were the variables analyzed and what type of variables were they really?
How do authors define "awareness"?
L211-212: With what criteria?
L235-281, 824: It is not a suitable design to draw inferences about this. I am struck by how naturally statements are made without clear analysis. It is really a shame that the authors did not use scales and an appropriate design to answer these questions. I suggest that the authors consult a statistician for future work.
Everything mentioned here is totally subjective and without test analysis, and it is not a clear objective of the study.
L899-900: This statement is not a conclusion that follows from this paper, and on the other hand, this paper is not needed to arrive at that statement, which is widely known...
Round 2
Reviewer 1 Report
Comments and Suggestions for Authors
Solid introduction in this draft
Typo - errors
L 142 missing period broaden the reach.
L 205 What language welfare language....
L 339 ...I suspect its [positive welfare] is this is a quote so may not be an oversite
L 394-395 intuition appears twice
L 621 who herds a heritage sheep breed
L 622 They showed
L625 awareness of positive welfare
L 637 dissemination of the
L 644 farmers feared the ---- phrase or word missing
L 648 lost the pop OR lost the plot - regional slang
L 735 remove (.) after changed
Reviewer 4 Report
Comments and Suggestions for Authors
I understand the effort that the authors put into the manuscript. Unfortunately, the responses, more specifically related to the design and statistical analysis, reinforce the major weaknesses that could not be improved by the authors. In addition, there are certain other concerns regarding some of the responses. Under these conditions, the article is not in a position to be accepted.